# Confronting the Negative Impact of Cigarette Smoking on Cancer Surgery

**Se-In Choe and Christian Finley \***

Department of Surgery, Division of Thoracic Surgery, McMaster University, Hamilton, ON L8N 4A6, Canada
\* Correspondence: finleyc@mcmaster.ca

**Abstract:** Smoking is a common health risk behavior that has substantial effects on perioperative risk and postoperative surgical outcomes. Current smoking is clearly linked to an increased risk of perioperative cardiovascular, pulmonary and wound healing complications. Accumulating evidence indicates that smoking cessation can reduce the higher perioperative complication risk that is observed in current smokers. In addition, continued smoking has a negative impact on the overall prognosis of cancer patients. Smoking cessation, on the other hand, can improve long-term outcomes after surgery. Smoking cessation services should be implemented in a comprehensive programmatic manner to ensure that all patients gain access to evidence-based care. Although the benefits of abstinence increase in proportion to the length of cessation, cessation should be recommended regardless of timing prior to surgery.

**Keywords:** surgery; smoking cessation; cancer

## 1. Introduction

Smoking tobacco is a common health risk behavior among the general adult population worldwide. It is the leading cause of preventable morbidity and premature mortality in the US [1,2]. It is also profoundly associated with negative health outcomes, including the risks of developing a wide range of different cancers. However, the effects of smoking combustible tobacco cigarettes on the preoperative risk and postoperative outcomes are less well appreciated [1]. This paper reviews these effects and advocates for the implementation of programmatic smoking cessation services and early preoperative efforts to help all patients who use tobacco to quit. Although the main focus of this paper is on cigarette smoking, the importance of smoking cessation also applies to other types of smoking, such as pipe and cigars.

There is growing evidence that smoking is associated with higher rates of both perioperative and postoperative complications. Current smoking exposes patients scheduled for surgery to a 20% increased risk of in-hospital mortality and a 40% increased risk of a major postoperative complication [3]. With a significant impact on postoperative course, such as wound healing, infection, anastomotic leaks, as well as cardiovascular and respiratory complications, smoking cessation may be effective in mitigating morbidity risks associated with active smoking and cancer surgery. Smoking is one of the most modifiable predictors of operative success. Perioperative smoking cessation should always be recommended to patients by surgeons because of the increased risk of postoperative complications from tobacco use.

## 2. Underlying Pathophysiology and Its Effects on Surgical Outcomes

Cigarette smoking is associated with a wide range of toxic effects on virtually every body system. Research has demonstrated that cellular damage from smoking is secondary to a variety of factors, including the release of free radicals, tissue hypoxia, compromised

immune cell function, and microvascular injury, leading to dysfunction and thrombogenesis [1,2]. These mechanisms decrease oxygen transport and increase the risk of a cardiovascular event [4]. Moreover, increased oxidative stress inhibits proper functioning of the immune system, which impairs wound healing and reduces the body's defense against infections [2,4]. More specifically, smoking impairs production of pro and anti-inflammatory cytokines responsible for regulating immune function, which in turn predispose patients who smoke to infections in the postoperative state [4]. All these mechanisms of cellular damage contribute to the complications observed in the perioperative period. Cigarette smoke also contains more than 4000 toxic compounds (more than 70 known carcinogens), some of which also cause impairment in wound healing [5]. In addition, nicotine, carbon monoxide, and hydrogen cyanide combine to decrease the amount of available oxygen and other substances required for the healing process and inflammatory response [4]. The synthesis of subcutaneous collagen is impaired in persons who smoke, which also contributes to poor wound healing [4,5]. Furthermore, there is evidence that smoking can directly impair osteoblast activity, which may impair wound healing and is a risk factor for non-union of long bones [4]. Smoking also has a negative impact on pulmonary function by increasing mucus secretion and decreasing muco-ciliary clearance, which can lead to an increase in infections and respiratory complications postoperatively [2,4].

## 3. Impact on Surgical Outcomes

Accumulating evidence indicates that smoking cessation can reduce the higher perioperative complication risk that is observed in current smokers and it also improves long-term outcomes after surgery [1,2]. In the medical literature, the impact of smoking on perioperative outcomes has been relatively well defined in numerous surgical specialties, including general surgery, cardiothoracic surgery, orthopedic surgery, plastic surgery, and neurosurgery [1]. Even secondhand smoke exposure has been shown to be a risk factor in pediatric surgical outcomes [1]. Active smoking is clearly linked to an increased risk of perioperative cardiovascular, pulmonary complications and wound healing complications, including wound infections and anastomotic dehiscence [1]. These complications result in longer hospital admissions, higher rates of ICU admissions, increased risk of requiring a secondary surgery and higher overall costs of care.

In the field of neurosurgery, smoking has been found to be associated with increased risk for intracranial aneurysm formation, aneurysmal subarachnoid hemorrhage and decreased bone healing and fusion after spinal procedures [1]. In head and neck cancer surgery, smoking cessation is effective in reducing wound complications following various surgical procedures, such as free flaps [4]. In a retrospective study by Kuri et al., preoperative smoking cessation longer than 3 weeks reduced the incidence of impaired wound healing among patients who underwent reconstructive head and neck cancer surgery, compared to the incidence observed in non-smokers [6]. The incidences (95% confidence intervals) of impaired wound healing in this study were as follows: 85.7% (73–97%) in smokers, 67.6% (52–83%) in late quitters, 55% (33–77%) in intermediate quitters, 59.1% (47–71%) in early quitters and 47.5% (32–63%) in nonsmokers [6] where late, intermediate and early quitters were defined as patients whose duration of smoking cessation was 8–21, 22–42 and 43 days or longer before their operation, respectively [6]. Interestingly, in a multivariate analysis, the authors concluded that the total amount of cigarette smoking (number of pack-years) did not have a significant impact on the association between smoking cessation and impaired wound healing [6]. The association of poor postoperative outcomes related to smoking is also observed in colorectal surgery [7]. In a retrospective study reviewing patients who underwent elective right hemicolectomies for colon cancer [7], smokers were found to have a higher rate of organ space infection, unplanned return to the operating room and risk of anastomotic leak. Smoking was also found to be an independent risk factor for wound complications, primary pulmonary complications, and anastomotic dehiscence. Fawcet et al. demonstrated that smoking was correlated with microvascular disease in resected colonic specimens, and interestingly, the diseased

specimens had statistically significant increased rates of anastomotic failures [8]. Similarly, in hepatobiliary surgery, smoking status has been found to be an independent risk factor for postoperative pancreatic fistula following pancreaticoduodenectomy for cancer [8]. In thoracic surgery, the risks of developing postoperative pulmonary complications, such as pneumonia and pneumothorax, are higher with smoking [9]. These complications are also associated with increased in-hospital mortality, intensive care unit admissions, longer hospital admission stays and worse long-term outcomes [9].

A recent meta-analysis was performed to study the effect of preoperative smoking and cessation on postoperative wound healing [10]. Patients who stopped smoking or non-smokers had significantly lower postoperative wound healing problems, and surgical site wound infection compared with smokers. Another systematic review confirmed that smokers have a higher risk of postoperative healing complications compared to non-smokers [11]. A similar trend was observed when comparing former smokers with non-smokers. In addition, perioperative smoking cessation significantly reduced rates of surgical site infections [11]. Certainly, postoperative wound infections are a huge burden to healthcare systems. A cost analysis study in Australia [12] showed that substantial short-term health and economic benefits were associated with reducing surgical site infections (SSI) just from achieving higher smoking cessation rates. The authors reported that a reduction in the estimated surgical smoking rate from 23.9% to 5% can allow overall estimated cost savings of 26.1M Australian Dollars (AUD) in annual hospitalization for SSI, respectively [12]. Although findings of this study are applicable in the Australian healthcare setting, the authors concluded that their methods can be applied to other similarly public funded healthcare systems, such as Canada [12].

For virtually every cancer site, active smoking has been shown to negatively impact each step of the surgical journey and result in negative outcomes for patients undergoing cancer surgery.

## 4. Impact on Surgical Cancer Prognosis

Ongoing smoking has an impact on overall prognosis of cancer patients, as well as impairing the success of other anti-cancer treatments [13,14]. Several in vivo studies have demonstrated that smoking could increase the biologic aggressiveness of tumor cells by promoting tumor cell proliferation, migration, invasion, metastasis, and angiogenesis [13,14]. In addition, smoking can affect drug disposition during treatment by influencing transcriptional and epigenetic regulation of cytochrome P450 enzymes [13,14].

Active smoking has been shown to be associated with higher rates of all-cause mortality in several different cancer types. In lung cancer, there is substantial data that persistent smoking in the preoperative period results in worse clinical outcomes, including postoperative morbidity and mortality. The effect of tobacco smoke, nicotine and the carcinogenic compounds in tobacco smoke can influence tumor growth, produce cellular damage and genetic mutations, cause immunosuppression, leading to tumor recurrence and the development of other comorbidities [15]. In addition, continued smoking can increase resistance to other treatments used in the treatment of lung cancer and increase the toxicity of treatment. Smoking has also been implicated in a wide range of adverse long-term outcomes among lung cancer patients, including a higher risk for second primary cancers and all-cause mortality [15]. Lung cancer surgery patients who are current smokers have an 86% increased risk for cancer recurrence and 2-fold decrease in 5-year survival compared with patients who quit smoking upon their initial diagnosis of lung cancer [9]. A prospective cohort study demonstrated that persistent smoking cessation following a diagnosis of lung cancer was associated with reduced overall survival, and progression-free survival and increased cancer-specific mortality [15]. The beneficial effects of smoking cessation were observed in all subgroups of patients with lung cancer in this study, including patients with early and advanced stage cancer, patients who have or not received chemotherapy or radiation, and mild to heavy smokers [15].

In head and neck squamous cell carcinoma, a systematic review has shown that smoking cessation is associated with improved outcomes, such as overall survival, recurrence rate and secondary primary tumor development [14]. In colorectal surgery, Amri et al. showed that active smoking was a stage-independent risk factor for hematogenous metastasis in colon cancer. This observation appeared to be unique to active smokers and was not observed in ex-smokers who had similar metastasis rates to non-smokers [16]. Smoking is also associated with a higher rate of breast cancer recurrence after surgery [7]. In a retrospective analysis, smoking was identified as an independent risk factor that affects breast cancer survival. Interestingly, in another retrospective study, Takada et al. showed that smoking may induce biological changes in recurrent breast cancer by increasing HER2 expression [17]. Other reports have described differences in the patterns of receptor expression in the original surgical specimens compared with recurrent tumor specimens [18]. This is important, as changes in receptor expression in breast cancer require a change in the choice of therapy. In genitourinary cancers, smoking is well known to be a preventable risk factor for both the development and the survival of bladder cancer, urothelial carcinoma, renal cell carcinoma, as well as prostate cancer. There is robust data that show the adverse effects of smoking on cancer outcomes after primary treatment for prostate cancer [19]. Some studies have also suggested a link between smoking and higher tumor grade, tumor volume and extracapsular extension during radical prostatectomy. A recent systematic review and meta-analysis showed that current smokers undergoing primary radical prostatectomy or radiotherapy had a significantly higher risk of recurrence (HR 1.40, 95% CI 1.18–1.66, $p < 0.001$), distant metastases (HR 2.51, 95% CI 1.80–3.51, $p < 0.001$) and cancer-specific mortality (HR 1.89, 95% CI 1.37–2.60, $p < 0.001$), when compared to non-smokers [19]. Although quantitative analyses on cumulative tobacco exposure and cessation were not feasible due to the heterogeneity of data in this study, the authors found two studies in their systematic review that showed a clinically significant advantage of smoking cessation of ten years or more in regard to biochemical recurrence of prostate cancer [19].

## 5. Optimal Timing of Smoking Cessation Prior to Surgery

Given the adverse effects of tobacco smoking on the surgical management of cancer and the negative impacts on survival, smoking cessation in the perioperative period should be routinely recommended to all currently smoking patients. Smoking cessation should be offered in a comprehensive programmatic manner to ensure that all patients can easily access high quality evidence-based care. Although the benefits of abstinence increase in proportion to the length of cessation, cessation should be recommended regardless of timing prior to surgery [3].

It remains unclear if there is an optimal duration of smoking cessation prior to surgery that minimizes the risks associated with smoking. Previous studies have demonstrated that most of the physiological changes from smoking are reversible to some extent, but the time period needed for significant improvement is likely to be between 6 and 8 weeks [5]. A randomized control trial from Denmark [5] randomized 120 patients to either smoking cessation 6–8 weeks before scheduled orthopedic surgery or to a control group. The overall complication rate in the smoking intervention group was 18% compared to 52% in the control group ($p = 0.0003$). Significant effects of the cessation intervention were observed for wound infections, cardiovascular complications, and the need for a second surgery. The investigators concluded that an effective smoking intervention 6–8 weeks prior to surgery reduces postoperative complications [5]. A systematic review and meta-analysis studied the relationship of short-term (<4 week) preoperative smoking cessation and postoperative complications [20]. Compared with current smokers, smokers who quit more than 4 and more than 8 weeks had lower risks of respiratory complications. Smokers who quit less than 2 or 2–4 weeks before surgery had similar risks of respiratory complications compared to smokers. For wound healing, the risk was lower in smokers who quit more than 3–4 weeks before surgery than in current smokers. In a recent study by Heiden et al., in patients with early-stage NSCLC, those who were able to stop smoking at least 3 weeks

before surgery had reduced postoperative mortality and complications [21]. For each extra week of abstinence before surgery, the authors found an 8.1% reduction rate in the odds of major complication or mortality cessation (OR for every additional week, 0.919; 95% CI 0.850–0.993, *p* = 0.03) [21].

## 6. Conclusions

The effect of smoking on preoperative and postoperative surgical outcomes cannot be underemphasized. It is the most modifiable factor in cancer surgery that is not fully leveraged. All surgeons should establish relationships with and refer to smoking cessation programs for all actively smoking patients for whom surgery is planned. The duration of abstinence from smoking appears to be important; therefore, efforts to help surgical candidates to stop smoking should be initiated at first contact and follow evidence-based interventions to optimize surgical outcomes.

**Author Contributions:** Conceptualization, C.F.; data curation, S.-I.C.; writing—original draft preparation, S.-I.C. and C.F.; writing—review and editing, S.-I.C. and C.F.; supervision, C.F.; project administration, C.F. All authors have read and agreed to the published version of the manuscript.

**Funding:** This research received no external funding.

**Institutional Review Board Statement:** Not applicable.

**Informed Consent Statement:** Not applicable.

**Data Availability Statement:** Not applicable.

**Conflicts of Interest:** The authors declare no conflict of interest.

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
