# Peer review of "Confronting the Negative Impact of Cigarette Smoking on Cancer Surgery"

_curroncol, doi:10.3390/curroncol29080463_

Round 1

Reviewer 1 Report

1. What is the main question addressed by the research?

Commentary:   Mitigating morbidity risks associated with active smoking and cancer surgery – Aim primarily to advocate for the implementation of programmatic smoking cessation services.

2. Do you consider the topic original or relevant in the field, and if
so, why?

Commentary – Not Original but Relevant:

There is robust evidence regarding the harmful effects of smoking on the human body.  It is clear that smoking cessation improves outcomes, yet despite clinical and cost advantages health systems remain slow to develop programmatic smoking cessation services.   From the perspective of the patient, the importance of educating the benefits of perioperative smoking cessation cannot be overstated.  Patients will often consider cessation as “too late”, when evidence clearly shows that there are significant benefits.

3. What does it add to the subject area compared with other published
material?

No new material but as a commentary it is concise and timely.  Referencing is up to date.

It may have been useful to insert a paragraph on effectiveness of smoking cessation aids – Champix, NRT in this setting.

4. What specific improvements could the authors consider regarding the
methodology?

Not applicable - Commentary

5. Are the conclusions consistent with the evidence and arguments
presented and do they address the main question posed?

As a commentary, it addresses the core premise and is consistent with the evidence and arguments presented

6. Are the references appropriate?  Yes

Several additional references could be used.  However, in this instance the Authors have achieved a good balance for a Commentary.

Author Response

We would like to thank the reviewer for the comments provided.

  1. What is the main question addressed by the research?

Commentary:   Mitigating morbidity risks associated with active smoking and cancer surgery – Aim primarily to advocate for the implementation of programmatic smoking cessation services.

  1. Do you consider the topic original or relevant in the field, and if
    so, why?

Commentary – Not Original but Relevant:

There is robust evidence regarding the harmful effects of smoking on the human body.  It is clear that smoking cessation improves outcomes, yet despite clinical and cost advantages health systems remain slow to develop programmatic smoking cessation services.   From the perspective of the patient, the importance of educating the benefits of perioperative smoking cessation cannot be overstated.  Patients will often consider cessation as “too late”, when evidence clearly shows that there are significant benefits.

  1. What does it add to the subject area compared with other published
    material?

No new material but as a commentary it is concise and timely.  Referencing is up to date.

It may have been useful to insert a paragraph on effectiveness of smoking cessation aids – Champix, NRT in this setting.

Thank you for the suggestion, but we believe that this was beyond the scope of our paper.

  1. What specific improvements could the authors consider regarding the
    methodology?

 Not applicable - Commentary

  1. Are the conclusions consistent with the evidence and arguments
    presented and do they address the main question posed?

As a commentary, it addresses the core premise and is consistent with the evidence and arguments presented

  1. Are the references appropriate? Yes

 Several additional references could be used.  However, in this instance the Authors have achieved a good balance for a Commentary.

Reviewer 2 Report

This manuscript is a commentary that provides a synopsis of why cigarette smoking cessation is important in cancer patients, with primary focus on cigarette smoking cessation prior to surgery.  The commentary is well organized, following a natural flow from the biological basis of the adverse effects of smoking combustible tobacco cigarettes on surgical outcomes, a consideration of the evidence of the impact of smoking combustible tobacco cigarettes on surgical outcomes for surgery considered broadly, the impact of persistent cigarette smoking on prognosis in cancer patients, and a consideration of the timing of cigarette smoking cessation before surgery.  This sequencing of topics is efficient for providing the reader with information to understand the key issues related to this topic.  Overall, this is a very important topic and the current version of the manuscript does a reasonable job of laying out the evidence supporting a referral to cigarette smoking cessation programs as a much needed early step following a cancer diagnosis.  Thus, the content of the manuscript rates highly for significance.

Despite these strengths, as noted below there are many ways the manuscript needs to be further strengthened.

Major Comments

1. Title: The current title reads “Confronting Negative Impact of Smoking on Cancer Surgery.”  In the title and throughout the manuscript, it will be important to refer specifically to at least “cigarette smoking” if not “combustible tobacco cigarette smoking.”   Although I will not provide minor technical edits in the body of the manuscript, the current title is also missing a “the” between “Confronting Negative” so a recommended revised title is “Confronting the Negative Impact of Cigarette Smoking on Cancer Surgery.”

2. Just to emphasize the importance of this point, this comment is to reiterate the importance of at least early on in the introductory text specifying that the term “smoking” is used to refer to smoking combustible tobacco cigarettes throughout.

3. It will be helpful to insert a sentence somewhere in the manuscript that while your focus has been on cigarette smoking, the importance of smoking cessation is likely to apply also to pipe and cigar smoking.

4. In the Abstract, (page) line 10, change “has an impact” to “has a negative impact”

5.  Introduction, par. 1, page line 20: Ultimately in the long-term we cannot prevent death, so change “mortality” to “premature mortality.”

6.  Introduction, par. 1, page line 2: The adverse health effects of cigarette smoking are so wide-ranging that hard to judge what is most notable, so change “, most notably” to “, including”

7.  A very important comment is that the referencing is inconsistent throughout.  For example, in Section 2 on underlying pathophysiology there are several statements of fact with no reference provided (e.g., last sentence in this section).  Also lacking is a reference that provides an overall comprehensive assessment of the health effects of smoking that could be used as a reference frequently.  Reference #1 is currently used but this is a 2013 article specific to neurological outcomes.  My recommendation is that the authors use the 2014 Report of the US Surgeon General as the primary reference for health effects.  Even though it is now 8 years old, it is currently the most thorough document on the overall adverse health effects of cigarette smoking.  It also includes coverage of the topic of smoking in cancer patients, which will be helpful for Section 4.  Also, the 2020 Report of the US Surgeon General covers the health benefits of smoking cessation and includes a section on smoking cessation after a cancer diagnosis that will provide a useful reference.

8.  In section 3, p. 2, page lines 75-81, there is a mistake in the information.  Based on the data presented in the second sentence, the first sentence needs to be re-written to clarify that a decrease in incidence is compared with “current smokers” not compared with “non-smokers” as the sentence currently states.

9.  In section 3, p. 3, page lines 110-111, there is a mistake in the information.  The sentence needs to be re-written to clarify that a decrease to 10% smoking prevalence will save $19.1M and a decrease to 5% smoking prevalence with save 

Author Response

We would like to thank the reviewer for all the comments.

Major Comments

  1. Title: The current title reads “Confronting Negative Impact of Smoking on Cancer Surgery.”  In the title and throughout the manuscript, it will be important to refer specifically to at least “cigarette smoking” if not “combustible tobacco cigarette smoking.”   Although I will not provide minor technical edits in the body of the manuscript, the current title is also missing a “the” between “Confronting Negative” so a recommended revised title is “Confronting the Negative Impact of Cigarette Smoking on Cancer Surgery.”

This is an important point – we have changed the current title as suggested.

  1. Just to emphasize the importance of this point, this comment is to reiterate the importance of at least early on in the introductory text specifying that the term “smoking” is used to refer to smoking combustible tobacco cigarettes throughout.

We have added this important comment in the introduction line 28-29

  1. It will be helpful to insert a sentence somewhere in the manuscript that while your focus has been on cigarette smoking, the importance of smoking cessation is likely to apply also to pipe and cigar smoking.

This was added in lines 31-33.

  1. In the Abstract, (page) line 10, change “has an impact” to “has a negative impact”

Changed in line 15

  1. Introduction, par. 1, page line 20: Ultimately in the long-term we cannot prevent death, so change “mortality” to “premature mortality.”

Changed in line 26

  1. Introduction, par. 1, page line 2: The adverse health effects of cigarette smoking are so wide-ranging that hard to judge what is most notable, so change “, most notably” to “, including”

Changed in line 27

  1. A very important comment is that the referencing is inconsistent throughout.  For example, in Section 2 on underlying pathophysiology there are several statements of fact with no reference provided (e.g., last sentence in this section).  Also lacking is a reference that provides an overall comprehensive assessment of the health effects of smoking that could be used as a reference frequently.  Reference #1 is currently used but this is a 2013 article specific to neurological outcomes.  My recommendation is that the authors use the 2014 Report of the US Surgeon General as the primary reference for health effects.  Even though it is now 8 years old, it is currently the most thorough document on the overall adverse health effects of cigarette smoking.  It also includes coverage of the topic of smoking in cancer patients, which will be helpful for Section 4.  Also, the 2020 Report of the US Surgeon General covers the health benefits of smoking cessation and includes a section on smoking cessation after a cancer diagnosis that will provide a useful reference.

This is an important point. References were added as suggested: lines 48-67. The 2014 report of the US Surgeon General was incorporated to the text.

  1. In section 3, p. 2, page lines 75-81, there is a mistake in the information.  Based on the data presented in the second sentence, the first sentence needs to be re-written to clarify that a decrease in incidence is compared with “current smokers” not compared with “non-smokers” as the sentence currently states.

The sentence was rewritten for clarity (lines 70-72).

  1. In section 3, p. 3, page lines 110-111, there is a mistake in the information.  The sentence needs to be re-written to clarify that a decrease to 10% smoking prevalence will save $19.1M and a decrease to 5% smoking prevalence with save 

Decrease to 10% saves $19.1M and 5% saves $26.1M respectively (as stated lines 119-120).

Reviewer 3 Report

Dear Editor,

Thank you for the opportunity to review the manuscript entitled “Confronting Negative Impact of Smoking on Cancer Surgery” by Choe and Finley. This commentary gives a very good and brief overview of the detrimental effects of smoking on perioperative and postoperative outcomes in patients undergoing surgery, with a particular focus on patients with cancer. The authors also briefly review the underlying pathophysiologic mechanisms for these effects and discuss how smoking cessation could improve surgical outcomes, as well as the overall and progression-free survival in patients with cancer.

The manuscript is well-written, and the topic is a very important one that is unfortunately overlooked in clinical practice and in research. I have the following comments that could help to improve the manuscript:

1.      The second paragraph of the introduction can be removed as it seems like a conclusion. Instead, the authors can focus more on the gaps in clinical practice and in research that made them think of writing this commentary. Particularly, they can add a couple of sentences on how prevalent is smoking among patients undergoing surgery, or among patients with cancer. Also, they can discuss the gaps in the current literature that made them think of writing this commentary.

2.      I suggest re-organizing the subheadings 2 and 3. Particularly, I suggest after the introduction section, the authors first describe the complications related to smoking (subheading: impact on surgical outcomes), and then they discuss the underlying pathophysiology for these effects (subheading: Underlying Pathophysiology and Its Effects on Surgical Outcomes). This can help the readers first know what the complications are and then understand how these complications can occur.

3.      I encourage the authors to include a subheading or a section on the “current gaps in clinical practice” to describe the prevalence of smoking among patients undergoing surgery or among patients with cancer. Also, this part can describe the current status of implementing smoking cessation programs among patients who undergo surgery or patients with cancer. This part can discuss whether there are efforts to support cancer patients to stop smoking and if yes, what is the success or the coverage of these efforts.

4.      I also encourage the authors to include a paragraph to discuss the gaps in current research around this topic. Particularly, do the authors feel there is a need for more research to focus on smoking cessation among patients with surgery or with cancer? Are there enough intervention studies or prospective cohort studies assessing the benefits of these programs among patients with cancer? Or is there a need for such studies?

5.      Please add a reference for line 66 on the impact of second-hand smoking on pediatric surgical outcomes.

6.      Lines 81-82: please define early, intermediate, and late quitters.

7.      Lines 115-117 need a reference. Otherwise, please make clear this is coming from the authors as a conclusion of what they described earlier.

8.      Line 170, there is a typo error: “cessation of 10 years of more” should be corrected as “cessation of 10 years or more”.

Author Response

We would first like to thank the reviewer for all the comments and suggestions.

The manuscript is well-written, and the topic is a very important one that is unfortunately overlooked in clinical practice and in research. I have the following comments that could help to improve the manuscript:

  1. The second paragraph of the introduction can be removed as it seems like a conclusion. Instead, the authors can focus more on the gaps in clinical practice and in research that made them think of writing this commentary. Particularly, they can add a couple of sentences on how prevalent is smoking among patients undergoing surgery, or among patients with cancer. Also, they can discuss the gaps in the current literature that made them think of writing this commentary.

Thank you for your suggestion, but our commentary is a brief overview on the impact of smoking cessation on surgical outcomes of cancer patients. It would be certainly be interesting to discuss about gaps in clinical practice/research, but we believe that it is somewhat beyond the scope our current review.

  1. I suggest re-organizing the subheadings 2 and 3. Particularly, I suggest after the introduction section, the authors first describe the complications related to smoking (subheading: impact on surgical outcomes), and then they discuss the underlying pathophysiology for these effects (subheading: Underlying Pathophysiology and Its Effects on Surgical Outcomes). This can help the readers first know what the complications are and then understand how these complications can occur.

We would like to thank the reviewer for the suggestion, however, we wanted to discuss the pathophysiology first and then discuss about the 2 major outcomes for patients undergoing cancer surgery: impact of smoking on surgical outcomes and subsequently on cancer prognosis.

  1. I encourage the authors to include a subheading or a section on the “current gaps in clinical practice” to describe the prevalence of smoking among patients undergoing surgery or among patients with cancer. Also, this part can describe the current status of implementing smoking cessation programs among patients who undergo surgery or patients with cancer. This part can discuss whether there are efforts to support cancer patients to stop smoking and if yes, what is the success or the coverage of these efforts.

This is an important point and we thank the reviewer for bringing it up, while, at the same time, we believe it is somewhat beyond the scope of our current brief commentary.

  1. I also encourage the authors to include a paragraph to discuss the gaps in current research around this topic. Particularly, do the authors feel there is a need for more research to focus on smoking cessation among patients with surgery or with cancer? Are there enough intervention studies or prospective cohort studies assessing the benefits of these programs among patients with cancer? Or is there a need for such studies?

On our literature review, there is certainly a lot of current research for smoking cessation among patients with cancer and those who are having surgery. We thank the reviewer for the suggestion provided, but this would certainly be a topic of another commentary/review paper.

  1. Please add a reference for line 66 on the impact of second-hand smoking on pediatric surgical outcomes.

Reference was added (line 76)

  1. Lines 81-82: please define early, intermediate, and late quitters.

The definition was added (lines 90-92).

  1. Lines 115-117 need a reference. Otherwise, please make clear this is coming from the authors as a conclusion of what they described earlier.

 References are in the text

  1. Line 170, there is a typo error: “cessation of 10 years of more” should be corrected as “cessation of 10 years or more”.

Changed in line 180

Round 2

Reviewer 2 Report

Overall the authors have attempted to respond to the comments provided, and as a consequence the manuscript has improved considerably.  Only a few comments remain.

1. The word "cigarette" is mis-spelled in the title.

2. lines 23-24, change "preventable premature morbidity and mortality" to "preventable morbidity and premature mortality"

3.  lines 128-130.  This text is still quite confusing to the reader.  Change to just one scenario "...reduction of estimated surgical smoking rate from 23.9% to 5% would result in cost savings of $26.1M in annual hospitalization for SSI."

4. line 157-158.  I am certain the text should read "Lung cancer surgery patients who are current smokers have an increased risk..." rather than the current "Lung cancer surgery patients have an increased risk..."

5.  lines 160-162: this sentence implies that smoking cessation is associated with uniformly worse health outcomes: reduced survival and increased mortality.  Is the text supposed to read "...demonstrated that persistent smoking following a diagnosis of lung cancer..." rather than "...demonstrated that smoking cessation following a diagnosis of lung cancer..."  If the sentence is accurate as it is written, it runs counter to the arguments provided and will require more in-depth explanation.

Author Response

We thank the reviewer for revising our manuscript in detail and providing important comments!

  1. The word "cigarette" is mis-spelled in the title.

Title spelling corrected. Thank you!

  1. lines 23-24, change "preventable premature morbidity and mortality" to "preventable morbidity and premature mortality"

This was revised in line 25.

  1. lines 128-130.  This text is still quite confusing to the reader.  Change to just one scenario "...reduction of estimated surgical smoking rate from 23.9% to 5% would result in cost savings of $26.1M in annual hospitalization for SSI."

This was revised as suggested in line 122.

  1. line 157-158.  I am certain the text should read "Lung cancer surgery patients who are current smokershave an increased risk..." rather than the current "Lung cancer surgery patients have an increased risk..."

This was revised in line 148-149.

  1. lines 160-162: this sentence implies that smoking cessation is associated with uniformly worse health outcomes: reduced survival and increased mortality.  Is the text supposed to read "...demonstrated that persistent smokingfollowing a diagnosis of lung cancer..." rather than "...demonstrated that smoking cessation following a diagnosis of lung cancer..."  If the sentence is accurate as it is written, it runs counter to the arguments provided and will require more in-depth explanation.

This was changed in line 151.